# A cost-effective breast cancer screening strategy for Urban China: Findings from a Shenzhen-based modeling study

Changqing Tu[1], Yuke Zhong[1], Huan Li[2], Haifeng Qi[2], Qian Lu[3], Xuesen He🔟[1]*

**1** Shenzhen Longgang Central Hospital, Shenzhen, China, **2** Shenzhen Maternity and Child Healthcare Hospital, Shenzhen, China, **3** Shenzhen Longgang District Center for Disease Control and Prevention, Shenzhen, China

* lgzxyytcq@163.com

## Abstract

### Background

Early detection through breast cancer screening significantly enhances survival rates and reduces mortality. However, financial constraints in low- and middle-income countries often limit the implementation of large-scale screening programs. This study evaluates the cost-effectiveness of a combined Clinical Breast Examination (CBE), Breast Ultrasound (BUS), and supplementary Mammography (MAM), screening strategy for women aged 35–65 in Shenzhen, China. It further identifies optimal screening protocols by analyzing variations in screening frequency, starting/ending ages, and long-term health outcomes.

### Methods

A Markov model was developed from a societal perspective to assess the lifetime cost-effectiveness of biennial (CBE+BUS)+MAM screening for women aged 35–65. A total of 27 strategies were simulated, varying screening frequency (annual, biennial, triennial), age at initiation (35, 40, 45), age at cessation (65, 69, 70), and modality combinations. Quality-Adjusted Life Years (QALYs) served as the primary health outcome metric. Incremental Cost-Utility Ratios (ICURs) were calculated, with one-way and second-order Monte Carlo sensitivity analyses conducted to evaluate parameter uncertainty.

### Results

Among 699,600 participants, 724 breast cancer cases were detected (detection rate: 103.49 per 100,000), with 88% diagnosed at early stages. The current Shenzhen strategy – biennial Clinical Breast Examination combined with Breast Ultrasound and supplementary Mammography ((CBE+BUS)+MAM/2year/35_65) – demonstrated an ICUR of 140,915 CNY/QALY compared to no screening, below one times

**Data availability statement:** The basic modeling parameters are fully presented in the paper's tables, and the original individual-level breast cancer screening data, after rigorous anonymization, have been deposited in the Zenodo repository (DOI: 10.5281/zenodo.17482406) and are freely accessible to researchers.

**Funding:** The author(s) received no specific funding for this work.

**Competing interests:** The authors have declared that no competing interests exist.

the per capita GDP (indicating cost-effectiveness). In various scenarios, while the (CBE + BUS)+MAM/3year/45_65 strategy had a lower ICUR (95,545 CNY/QALY), the ICUR for the current strategy versus this alternative was 518,121 CNY/QALY, still below the willingness-to-pay threshold of 537,000 CNY (three times GDP). Second-order Monte Carlo simulations confirmed the robustness of the current strategy in 76% of scenarios.

## Conclusion

The (CBE + BUS) +MAM/2year/35_65 strategy was identified as the optimal choice among 27 alternatives, providing a cost-effective balance between early detection and resource efficiency. This evidence solidifies its use and offers a strategic framework for allocating public health resources in Shenzhen and comparable urban settings.

## Introduction

Breast cancer is the most prevalent malignancy among women globally, accounting for 11.6% of new cancer cases and 6.9% of cancer-related deaths [1]. In China, it remains a critical public health challenge, with an incidence rate of 51.17 per 100,000 women and a mortality rate of 10.86 per 100,000 [2]. The economic burden is substantial: in the U.S., treatment costs reached 26.2 billion USD, representing 14% of the total cancer treatment expenditure [3]. In China, direct medical expenditures in Liaoning Province totaled CNY 830.19 million (USD 122.96 million), accounting for 9.9% of the total cancer-associated health expenditure [4].

Early detection via screening is pivotal for reducing mortality. The U.S. National Cancer Institute's SEER database highlights a stark disparity in 5-year relative survival rates: nearly 99% for stage I versus 31% for stage IV disease [5]. Mammography, widely used in high-income countries [6], has limited sensitivity in dense breast tissue-a common trait among Asian women [7]. Ultrasound, being non-invasive and radiation-free, serves as a complementary method that improves detection rates in younger women with dense breast tissue [8].

Determining optimal screening protocols-age ranges, frequency, and method combinations-remains complex. Many high-income countries delay initiating screening for women aged 40–49 due to insufficient mortality reduction evidence [9–13], while biennial mammography for 50–69-year-olds is standard in Europe [14]. In China, screening strategies vary regionally: the Chinese Cancer Society recommends ultrasound for 40–65-year-olds, whereas the 2019 national strategy proposes biennial mammography for 45–70-year-olds [15]. However, these approaches have yielded suboptimal outcomes. The China National Breast Cancer Screening Project (CNBCSP) primarily employs clinical breast examination (CBE) for women aged 35–69, followed by imaging evaluation for suspected abnormalities [15].

The cost-effectiveness of breast cancer screening in China is highly context-dependent, with outcomes significantly influenced by screening modality, target

population, regional socioeconomic conditions, and local willingness-to-pay thresholds. This variability is particularly evident in evaluations of single-modality screening. For example, while some national-level analyses suggest mammography may be cost-effective for the general population [16], studies in specific urban settings such as Urumqi identified breast ultrasound as the more economically efficient option in that particular context [17, 18]. Furthermore, risk stratification reveals substantial economic differences: evidence indicates that mammography is generally not cost-effective for average-risk women but can be economically viable for high-risk subgroups [19]. Combined screening strategies have also demonstrated favorable cost-effectiveness in certain urban populations. Research in Chinese urban areas shows that clinical breast examination (CBE) combined with breast ultrasound (BUS), supplemented by mammography (MAM), is particularly cost-effective for women aged 50–59 [8]. However, the same CBE + BUS strategy tends not to be cost-effective in rural regions [18], underscoring the substantial urban–rural disparities in screening economics. In summary, there is no universally "optimal" breast cancer screening strategy for China; rather, "context-specific" and "risk-adapted" approaches are essential. Shenzhen, as a highly developed megacity with concentrated healthcare resources, exhibits unique female demographic and breast cancer epidemiologic characteristics. Nevertheless, there is currently a lack of large-scale, local empirical studies evaluating the cost-effectiveness of a combined (CBE + BUS)+MAM strategy for women across the broad 35–65 age range. This study aims to fill this gap by developing a decision-analytic model to establish a scientifically sound and economically efficient localized breast cancer screening strategy for Shenzhen.

This study evaluates the cost-effectiveness of a (CBE + BUS) +MAM (Clinical Breast Examination, Breast Ultrasound, and supplementary Mammography) screening program for women aged 35–65 in Shenzhen, China. By simulating 27 strategies (S2 Table) varying screening frequency (annual, biennial, triennial) and age ranges (35/40/45–65/69/74), we aim to identify a precision screening model. A Markov decision model from a societal perspective uses Quality-Adjusted Life Years (QALYs) as the primary health outcome metric. Incremental Cost-Utility Ratios (ICURs) and sensitivity analyses will assess parameter uncertainty, providing evidence for policy formulation in urban China.

## Methods

### Screening protocol and clinical pathway

This study is a retrospective study based on data from the Breast Cancer Screening Project in Shenzhen. The data used in this study cover the period from 01/01/2021 to 31/12/2023. These data were accessed on 09/01/2024 for cost-effectiveness analysis. All data were fully anonymized before our analysis. The evaluated strategy consisted of an initial Clinical Breast Examination (CBE) combined with Breast Ultrasound (BUS). Participants with positive findings (BI-RADS IV/V) proceeded directly to biopsy for pathological diagnosis. Those with suspicious or inconclusive results (BI-RADS 0/III) underwent supplementary Mammography (MAM) for further assessment, with positive MAM results leading to biopsy. This integrated (CBE + BUS) +MAM clinical pathway is detailed in (Fig 1). (A complete description of the screening protocol and clinical pathway is provided in S1 Appendix A.1).

### Analytical framework: The markov model and cohort simulation

A state-transition Markov model was developed using TreeAge Pro 2011 [20] to evaluate the lifetime cost-effectiveness of breast cancer screening strategies. The model structure and parameters were adapted from Wong et al. [21], simulating the natural history of breast cancer as a cohort progresses through mutually exclusive health states, including Healthy, Ductal Carcinoma in Situ (DCIS), Breast Cancer (Stages I-IV), and Death(Fig 2).

The model simulated a closed cohort of 100,000 women, all starting in the "Healthy" state at age 35. The simulation was run over 50 annual cycles, following the cohort until age 85 to capture the majority of the population's lifespan and the significant increase in breast cancer incidence with age. In each cycle, individuals could transition between health states based on predefined, age-dependent probabilities.

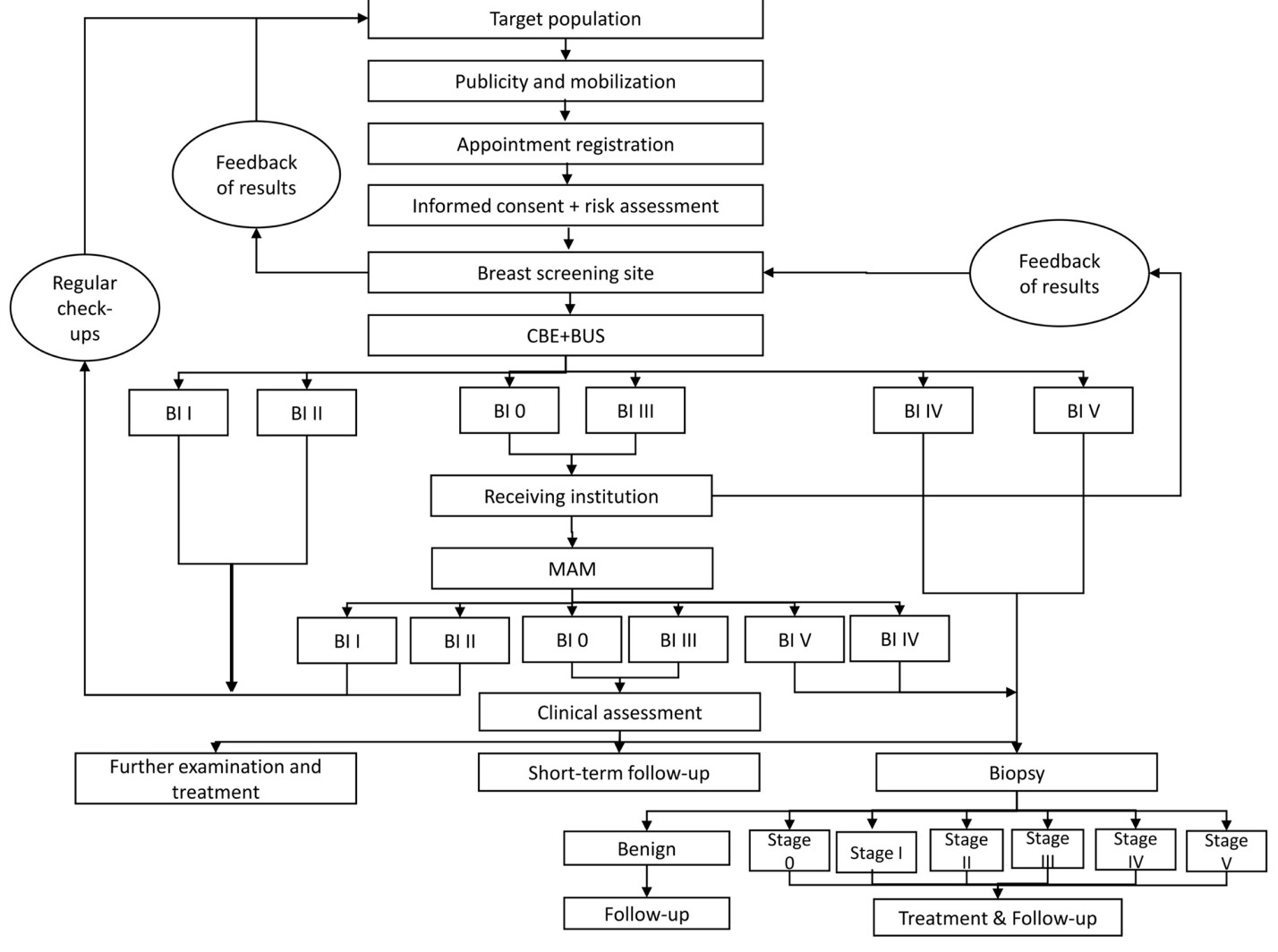

**Fig 1. Screening and Clinical Assessment Pathway for the (CBE+BUS) +MAM Strategy.**

We evaluated a total of 27 screening strategies (S2 Table), which varied systematically by starting age (35, 40, 45), ending age (65, 69, 74), and screening frequency (annual, biennial, triennial). For each strategy, the model tracked the accumulation of lifetime costs and Quality-Adjusted Life Years (QALYs) per woman. (The complete analytical framework is detailed in S1 Appendix A.2).

## Model parameters

The model required inputs for Initial probabilities, transition probabilities, costs, utilities, and screening performance. A comprehensive summary of key parameters and their sources is provided below:

Initial probabilities were based on actual data. Transition Probabilities: Probabilities for disease progression (e.g., from Stage I to Stage II) were derived from Wong et al. [21] and are detailed in Table 1. Epidemiological Data: Age-specific breast cancer incidence rates for urban Chinese women were sourced from the National Cancer Center of China [22] (Table 2). Age-specific all-cause and non-breast-cancer mortality rates were obtained

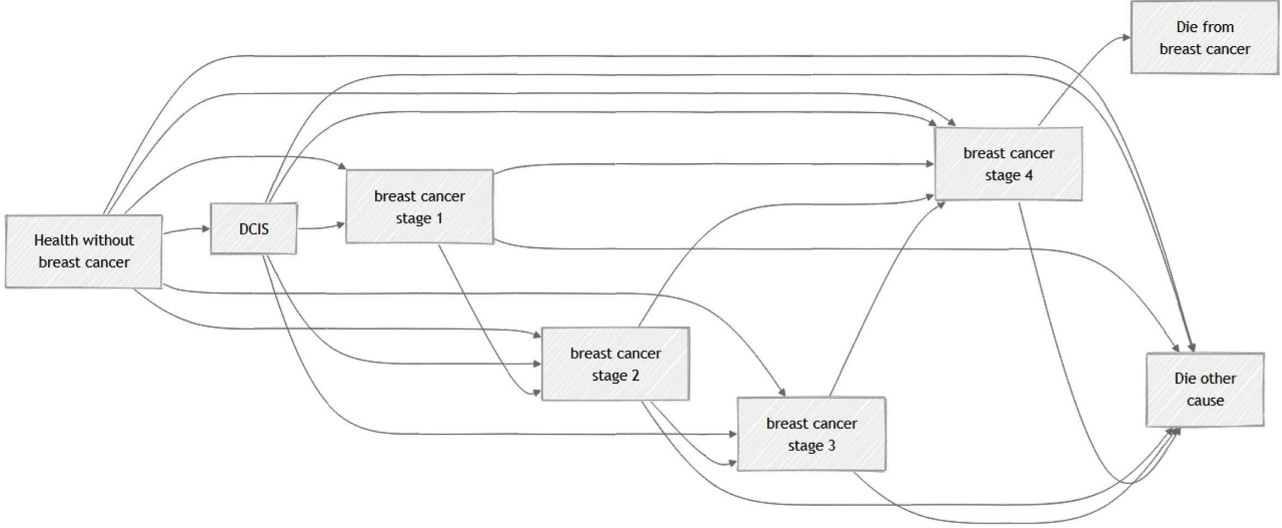

**Fig 2. Simplified flowchart.**

**Table 1. Key Transition and Risk Parameters in the Markov Model for Breast Cancer Screening.**

| Item | Value | Reference |
|---|---|---|
| Average annual progression probability of breast cancer staging | | |
| Stage I–Stage II | 0.06 | [18] |
| Stage II–Stage III | 0.11 | |
| Stage I–Stage IV | 0.01 | [21] |
| Stage II–Stage IV | 0.08 | |
| Stage III–Stage IV | 0.21 | |
| Breast cancer mortality after one year of treatment | | |
| Stage I | 0.002 | |
| Stage II | 0.016 | |
| Stage III | 0.039 | |
| Stage IV | 0.2331 | |
| Relative risk (RR) of invasive cancer from DICS | 2.02 | |

RR: Relative Risk of progressing from DCIS to invasive breast cancer, as derived from [21]. A value of 2.02 indicates that women with DCIS have a 2.02 times higher risk of developing invasive cancer compared to those without DCIS.

from the China Health Statistics Yearbook [23] (Table 3). The stage distribution of cancer at diagnosis under screening and no-screening scenarios, used for model calibration, is provided in Table 4. Screening Performance: The sensitivity and specificity for the combined (CBE + BUS)+MAM strategy were calculated as weighted averages from studies on previous studies [24, 25], as shown in Table 8.Cost Parameters: All costs, including screening, biopsy, and stage-specific treatment costs (from a societal perspective), were obtained from the Shenzhen screening program and published Chinese studies, adjusted to 2023 values (Tables 5,6).Health Utility Weights: utility values for different health states, essential for QALY calculation, were adopted from Wong et al. and are presented in Table 7 [21, 26].

**Table 2. Age-specific incidence of breast cancer.**

| Age | Urban China | Reference |
|---|---|---|
| 35-39 | 0.000342 | [22] |
| 40-44 | 0.000648 | |
| 45-49 | 0.000857 | |
| 50-54 | 0.001052 | |
| 55-59 | 0.001077 | |
| 60-64 | 0.001086 | |
| 65-69 | 0.000987 | |
| 70-74 | 0.000925 | |
| 75-79 | 0.000834 | |
| >=80 | 0.000637 | |

**Table 3. State transition probability parameters of Markov model for breast cancer screening, Transition probabilities of breast cancer.**

| Age | All-cause mortality | Breast cancer mortality | Normal death | Reference |
|---|---|---|---|---|
| 35-39 | 0.00052 | 3.76E-05 | 0.000482 | [23] |
| 40-44 | 0.0008 | 5.26E-05 | 0.000747 | |
| 45-49 | 0.00125 | 8.33E-05 | 0.001167 | |
| 50-54 | 0.00178 | 0.000113 | 0.001667 | |
| 55-59 | 0.00331 | 0.000162 | 0.003148 | |
| 60-64 | 0.00506 | 0.000184 | 0.004876 | |
| 65-69 | 0.0088 | 0.000228 | 0.008572 | |
| 70-74 | 0.01581 | 0.000231 | 0.015579 | |
| 75-79 | 0.02955 | 0.000265 | 0.029285 | |
| 80-84 | 0.05208 | 0.000376 | 0.051704 | |
| >=85 | 0.17321 | 0.000562 | 0.172646 | |

**Table 4. Stage distribution probability parameter table of Markov model for breast cancer screening.**

| Item | Screening | No screening | Reference |
|---|---|---|---|
| DCIS | 0.1119 | 0.0588 | Screening: Shenzhen Municipal Health Commission, Breast Cancer Screening Program (2021–2023) |
| Stage I | 0.3785 | 0.2652 | No screening: Model calibration |
| Stage II | 0.3881 | 0.457 | |
| Stage III | 0.1008 | 0.1529 | |
| Stage IV | 0.0207 | 0.0661 | |

## Model calibration, validation and outcomes

We internally calibrated the model to match the observed distribution of breast cancer stages at diagnosis in an unscreened Chinese population. For external validation, the model's predicted age-specific incidence rates were compared against independent national cancer registry data. The goodness-of-fit was assessed using a Chi-square test, which compared the observed number of cases from the registry with the model-predicted numbers across different age strata. The test showed no significant difference (Chi-square test, $x^2 = 2.366$, $p > 0.05$), supporting the model's validity (S1 Fig). (full calibration and validation procedures are detailed in S1 Appendix A.3).

**Table 5. Cost parameter estimates for screening program.**

| Screening cost | Cost Parameters (CNY) | Source |
|---|---|---|
| CBE | 33 | Shenzhen Municipal Health Commission, Breast Cancer Screening Program (2021–2023) |
| MAM | 249.33 | |
| US | 120 | |
| Histopathological Diagnosis | 184 | |

**Table 6. Medical costs (CNY).**

| TNM staging | Screening | No screening | Outpatient expenses | Direct non-medical costs |
|---|---|---|---|---|
| DCIS | 11432(7089-13186) | 13129(5742-19196) | 2710(739-6160) | 18108(3942-34626) |
| Stage I | 13545(12758-15751) | 18103(13730-23692) | 4466(2242-29262) | 12587(2562-26248) |
| StagII | 13104(11499-14992) | 19798(14785-27026) | 5113(1249-23595) | 8497(2464-26140) |
| StagIII | 14397(12774-16019) | 21138(16485-26791) | 3696(961-44356) | 6637(492-26090) |
| StagIV | 13292(8990-17593) | 29174(15092-37069) | 4928(1971-55445) | 11491(3696-15637) |

Note: The highest and lowest costs are shown in parentheses.

**Table 7. Health utility value.**

| Staging | Value | Reference |
|---|---|---|
| Health | 1 | [21] |
| DCIS | 0.95 | |
| Stage I | 0.9 | |
| StageII | 0.8 | |
| Stage III | 0.7 | |
| Stage IV | 0.3 | |

## Willingness-to-Pay threshold

This study uses the Incremental Cost-Utility Ratio (ICUR) as the evaluation standard, which measures the cost per Quality-Adjusted Life Year (QALY) gained. To determine the cost-effectiveness of screening strategies, we compare the ICUR with a predetermined Willingness-to-Pay (WTP) threshold. The evaluation criteria used include the World Health Organization's (WHO) recommendation for developing countries, which is three times the per capita GDP per QALY. According to the data from the Shenzhen Statistical Bureau, the per capita GDP of Shenzhen for the years 2021–2023 was 179,000 CNY. Therefore, three times this amount would be 537,000 CNY. An ICUR below 179,000 CNY/QALY indicates that breast cancer screening for women in Shenzhen is highly cost-effective. An ICUR below 537,000 CNY is considered cost-effective [27].

## Cost-utility analysis

The primary outcomes for the cost-effectiveness analysis were lifetime costs (from a societal perspective) and Quality-Adjusted Life Years (QALYs), both discounted at an annual rate of 3%. The Incremental Cost-Utility Ratios (ICURs) was calculated, and strategies were compared on the cost-effectiveness frontier after excluding dominated strategies, using a willingness-to-pay threshold of three times the per capita GDP of Shenzhen (537,000 CNY/QALY). (The detailed steps for ICUR calculation and dominance removal are provided in S1 Appendix A.4). A full list and detailed description of all 27

screening strategies evaluated in this study (varying by starting age, ending age, and screening frequency) are provided in S2 Table.

### Sensitivity analysis

Sensitivity analysis was conducted to evaluate the robustness of the model outcomes to parameter uncertainty. We performed both deterministic (one-way) and probabilistic analyses.

A probabilistic sensitivity analysis was performed using 1,000 Monte Carlo simulations to assess the joint impact of parameter uncertainty. Input parameters were assigned appropriate probability distributions [28, 29], with values randomly sampled in each iteration. The complete specifications for all distributions are detailed in S3–S6 Tables, (which correspond to the base-case values in Tables 5-8). A full methodological description is provided in S1 Appendix, Section A.5.

### Detailed methodological description

For technical details and a complete methodological description, please refer to the S1 Appendix (Sections A.1-A.5).

### Ethics statement

This study is a retrospective study based on data from the 2021–2023 Breast Cancer Screening Project. The Research Ethics Committee of Shenzhen Maternal and Child Health Hospital has approved the study with reference number: SFYLS [2024]008. Informed consent from participants is not required as the data were collected retrospectively and anonymized prior to receipt and analysis.

## Results

### Breast Cancer Screening Situation

In Shenzhen City, the breast cancer screening adopts the strategy of (CBE + BUS)+MAM/2year/35_65, and a total of 699,600 people were screened in this round, with 159,927 categorized as BI-RADS 0 or III (22.9%) and 163,437 as BI-RADS IV or higher (2.3%). A total of 724 breast cancer cases were detected, yielding a detection rate of 103.49 per 100,000 women and an early diagnosis rate of 88%. The age distribution showed the highest proportion of breast cancer cases in the 40–44 age group (24.03%), followed by the 45–49 and 50–54 age groups (22.24% and 20.86%, respectively), with a notable proportion in the 35–39 age group (12.98%) (Table 9).

### Cost-utility analysis

We utilized a Markov model to evaluate the cost-utility of the current screening program (CBE + BUS)+MAM/2year/35_65 and simulated 27 alternative strategies (S2 Table), employing the dominance strategy selection method (as detailed in the Cost-Utility Analysis section of the methodology), to select a more appropriate screening strategy for implementation in Shenzhen. The model's output encompasses the costs, utilities, and average cost-utility ratios of the screening strategies, as well as the incremental costs, incremental utilities, and Incremental Cost-Utility Ratios derived from the comparison

**Table 8. Sensitivity and specificity parameters of breast cancer screening protocols.**

| Screening strategy | Sensitivity | Specificity | Reference |
|---|---|---|---|
| (CBE + BUS)+MAM | 0.851(0.758-0.944) | 0.967(0.963-0.971) | [30] |

Note: The values in parentheses represent the range from the minimum to the maximum for each parameter.

**Table 9. Detection of different stages of breast cancer in different age groups (per 100, 000 women).**

| Age group | Screening population | Breast cancer cases | Disc | I | II | III | IV | Ratio(%) | Incidence rate per 100,000 women |
|---|---|---|---|---|---|---|---|---|---|
| total | 699600 | 724 | 81 | 274 | 281 | 73 | 15 | 100.0 | 103 |
| 35 ~ 39 | 229772 | 94 | 13 | 36 | 37 | 5 | 3 | 12.98 | 41 |
| 40 ~ 44 | 169872 | 174 | 18 | 64 | 75 | 17 | 0 | 24.03 | 102 |
| 45 ~ 49 | 128706 | 161 | 26 | 55 | 56 | 20 | 4 | 22.24 | 125 |
| 50 ~ 54 | 93348 | 151 | 12 | 57 | 59 | 19 | 4 | 20.86 | 162 |
| 55 ~ 59 | 55079 | 92 | 5 | 42 | 38 | 5 | 2 | 12.71 | 167 |
| 60 ~ 65 | 20607 | 41 | 5 | 18 | 13 | 5 | 0 | 5.66 | 199 |

Abbreviations: Disc, Invasive Ductal Carcinoma.

Note: This table presents the number of breast cancer cases detected at different stages across various age groups, alongside the corresponding incidence rates per 100,000 women.

between no screening and various screening options. S1 Table displays the metrics for each screening strategy at the individual level (S1 Table).

In Shenzhen, China, the current breast cancer screening program for women aged 35–65, which includes biennial Clinical Breast Examination (CBE) combined with Breast Ultrasound (BUS) and Mammography (MAM), is cost-effective. At the individual level, the total cost for the (CBE+BUS)+MAM/2year/35_65 screening program is approximately 5,592.34 CNY, and the average cost to gain 24.215 QALYs is about 394.73 CNY. The Incremental Cost-Utility Ratio is 140,915 CNY/QALY, which is below one times the per capita GDP (179,000 CNY), demonstrating significant cost-utility.

Through the simulation of 27 different strategies, we have identified 5 dominant strategies (as detailed in the Cost-Utility Analysis section of the methodology). The results of the screening are presented in S1 Table, and Fig 3 illustrates the relationship between cost utility and cost distribution among different strategies within the breast cancer screening program. Strategies in the upper left region of the graph are categorized as Dominated Strategies, while the Undominated Strategies in the lower right corner are (CBE+BUS)+MAM/3year/45_65, (CBE+BUS)+MAM/1year/40_65, (CBE+BUS)+MAM/1year/35_65, (CBE+BUS)+MAM/1year/35_69, and (CBE+BUS)+MAM/1year/35_74. Their Incremental Cost-Utility Ratios (ICURs) are 95,544.73, 649,251.82, 723,837.43, 1,236,492.02 and 3,214,628.11 CNY/QALY, respectively. The (CBE+BUS)+MAM/3year/45_65 screening strategy has the lowest additional cost per QALY gained. Using the World Health Organization's recommended threshold of three times the per capita GDP as a benchmark, this is significantly below the Willingness to Pay (WTP) threshold, making it the cost-effective strategy. The other four Undominated Strategies have ICURs exceeding the WTP threshold and are not cost-effective.

Comparing the currently implemented screening strategy, (CBE+BUS)+MAM/2year/35_65, with the simulated optimal strategy, (CBE+BUS)+MAM/3year/45_65, the ICUR is 518,120.51 CNY/QALY, which is below the predetermined WTP threshold of 537,000 CNY (three times the per capita GDP), suggesting that the (CBE+BUS)+MAM/2year/35_65 strategy still retains cost-effectiveness.

## Sensitivity analysis

To analyze the impact of fluctuations in model parameters within their value ranges on model outcomes, a sensitivity analysis was conducted on two cost-effective breast cancer screening strategies ((CBE+BUS)+MAM/3year/45_65 and (CBE+BUS)+MAM/2year/35_65). The results presented in Fig 4 and Fig 5 allow us to observe the effects of parameter variations on the expected value of the Incremental Cost-Utility Ratio (ICUR) for each screening strategy. In the sensitivity analysis of the (CBE+BUS)+MAM/3year/45_65 and (CBE+BUS)+MAM/2year/35_65 screening strategies, it was found that the discount rate has the most significant impact on the model simulation results. The effects of other model

**Fig 3. Cost-utility scatter plot of breast cancer screening programs.** This figure presents the cost-utility scatter plot of the 27 simulated screening strategies. The vertical axis represents the total cost per person, and the horizontal axis represents the effectiveness in Quality-Adjusted Life Years (QALYs) per person. Strategies that are less effective and more costly than a combination of other strategies (i.e., dominated strategies) are plotted primarily in the top-left area of the graph relative to the cost-effectiveness frontier. The cost-effectiveness frontier, formed by the undominated strategies, runs from the bottom-left to the top-right. The undominated strategies are explicitly labeled in the figure and are, in order of increasing effectiveness: (CBE+BUS)+MAM/3year/45_65, (CBE+BUS)+MAM/1year/40_65, (CBE+BUS)+MAM/1year/35_65, (CBE+BUS)+MAM/1year/35_69, and (CBE+BUS)+MAM/1year/35_74. Their respective Incremental Cost-Utility Ratios (ICURs) are presented in the text.

parameters on the (CBE+BUS)+MAM/2year/35_65 simulation results are listed in descending order as follows: utility value of Stage IV, cost of ultrasound screening, cost of physical examination, cost of follow-up for Stage IV, sensitivity, cost of mammography, etc. This means that among these parameters, the discount rate, the technical cost of screening, and the cost of follow-up have the greatest impact on ICUR. The impact of these parameters on the (CBE+BUS)+MAM/3year/45_65 model is essentially consistent. It is worth noting that no matter how the model parameters of the (CBE+BUS)+MAM/2year/35_65 and (CBE+BUS)+MAM/3year/45_65 screening strategies fluctuate within their value ranges, their corresponding ICUR is always lower than the WTP threshold. Changes in parameters do not significantly alter ICUR values, indicating that the (CBE+BUS)+MAM/2year/35_65 and (CBE+BUS)+MAM/3year/45_65 screening strategies are not sensitive to fluctuations in model parameters.

Compared to the (CBE+BUS)+MAM/3year/45_65 screening strategy, the (CBE+BUS)+MAM/2year/35_65 screening strategy is overly sensitive to these parameters (Fig 6). In the paired sensitivity analysis of (CBE+BUS)+MAM/2year/35_65 and (CBE+BUS)+MAM/3year/45_65, the parameters that have a significant impact on the model are as

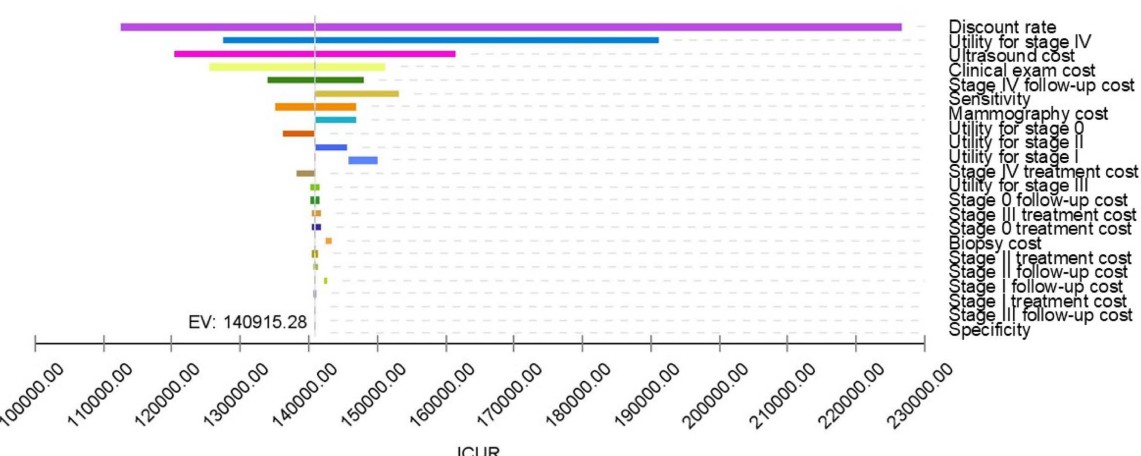

**Fig 4. Tornado diagram of one-way sensitivity analysis for (CBE + BUS) +MAM/2year/35-65 breast cancer screening vs no screening.**

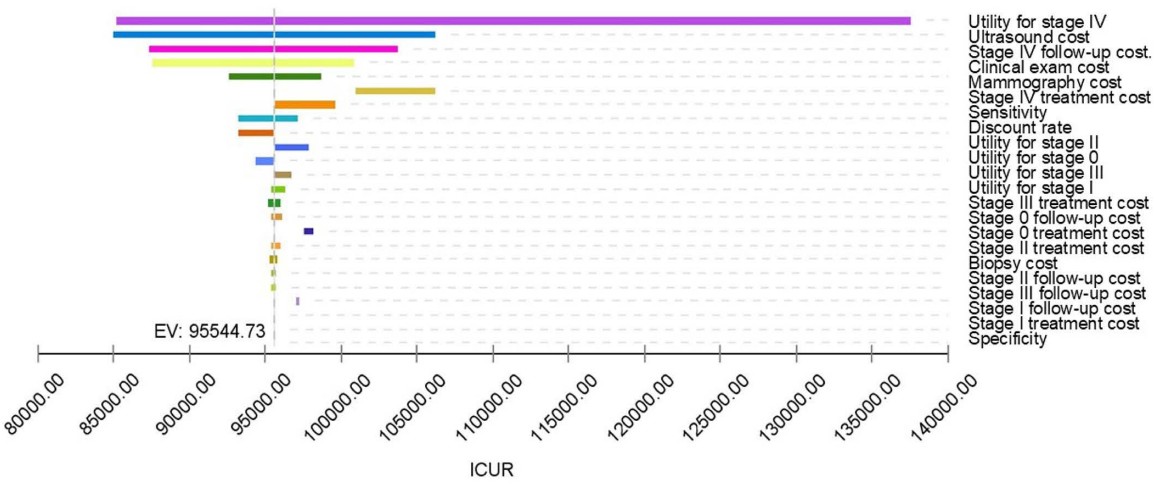

**Fig 5. Tornado diagram of one-way sensitivity analysis for (CBE + BUS) +MAM/3year/45_65 breast cancer screening vs no screening.**

follows: sensitivity, cost of ultrasound screening, utility of Stage I, cost of clinical examination, utility value of Stage 0, cost of mammography, discount rate, etc. Variations in these parameters can lead to ICUR values that approach or exceed the WTP threshold, and the uncertainty of these parameters could potentially affect the stability of the model.

This study utilized second-order Monte Carlo simulation with 1,000 runs to assess the cost-effectiveness of the (CBE + BUS)+MAM/2year/35_65 and (CBE + BUS)+MAM/3year/45_65 screening strategies compared to the no-screening option, resulting in a 95% confidence interval ICUR scatter plot (Fig 7). For the (CBE + BUS) +MAM/2year/35_65 screening strategy, when the willingness-to-pay (WTP) threshold is set at 537,000 CNY, 100% of the ICUR values fall below this

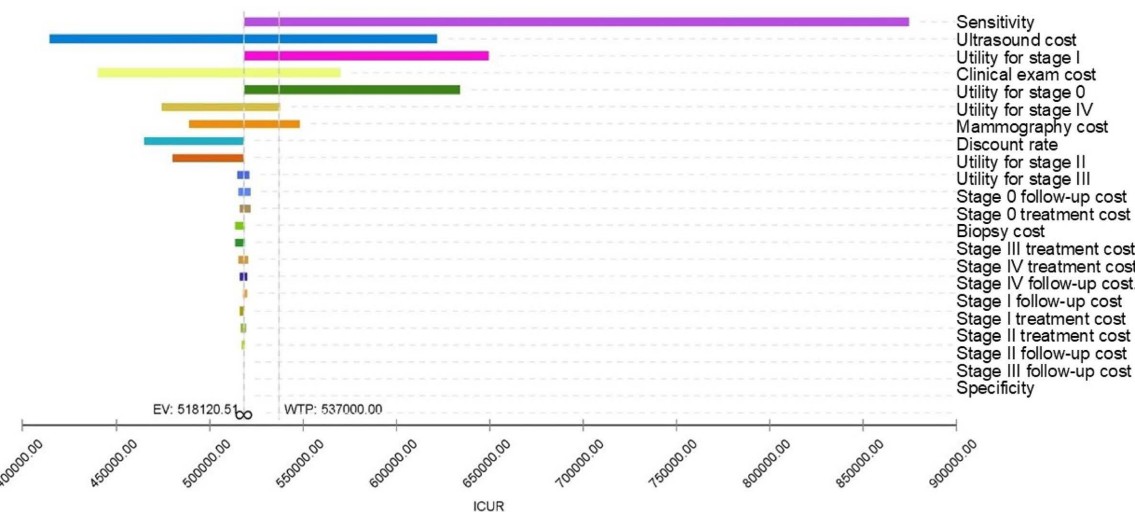

**Fig 6. Tornado diagram of one-way sensitivity analysis for (CBE + BUS) +MAM/3year/45_65 breast cancer screening vs (CBE + BUS)+MAM/2year/35-65.**

threshold line. This means that in 100% of the simulated scenarios, the (CBE + BUS) +MAM/2year/35_65 screening strategy is cost-effective compared to no screening. For the (CBE + BUS) +MAM/3year/45_65 screening strategy, under the same WTP threshold of 537,000 CNY, 100% of the ICUR values fall below the threshold line. This indicates that in 100% of the simulated scenarios, the (CBE + BUS) +MAM/3year/45_65 screening strategy has a cost-effectiveness advantage over no screening. Overall, the probability that both screening strategies have cost-effectiveness compared to no screening exceeds 50%, suggesting that the simulation results of the model are relatively stable.

For the paired comparison between the (CBE + BUS)+MAM/3year/45_65 and (CBE + BUS)+MAM/2year/35_65 screening strategies, in 76% of the simulated scenarios, the (CBE + BUS)+MAM/2year/35_65 screening strategy was more cost-effective than the (CBE + BUS)+MAM/3year/45_65 screening strategy (Fig 8).

## Discussion

The biennial (CBE + BUS)+MAM breast cancer screening program for women aged 35–65 in Shenzhen has been implemented for many years, yet its effectiveness has not been comprehensively evaluated. In this screening activity, a total of 699,600 eligible women were examined, with 724 cases of breast cancer detected. BI-RADS 0 and BI-RADS III accounted for 159,927 individuals, representing 22.9% of the screened population. Among these, ultrasound results indicating BI-RADS IV or higher were found in 163,437 individuals, which is 2.3%. This rate is higher than the provincial average [31]. After histological examination, the detection rate of breast cancer was 103.49 per 100,000 people, and the early diagnosis rate reached 88%. These figures significantly exceed the national average detection rate for urban areas (56 per 100,000) reported in 2016 [24]. This agrees with a study that found incidence higher in economically developed cities or eastern China [32]. The high detection and early diagnosis rates in Shenzhen's case may therefore be attributed to the effectiveness of biennial screening in ensuring early detection, supported by the city's high-quality medical resources, including qualified physicians and advanced screening equipment.

While the high detection and early diagnosis rates affirm the clinical value of the ongoing screening program, a critical policy question remains: Is the current strategy the most efficient use of limited healthcare resources, especially

A

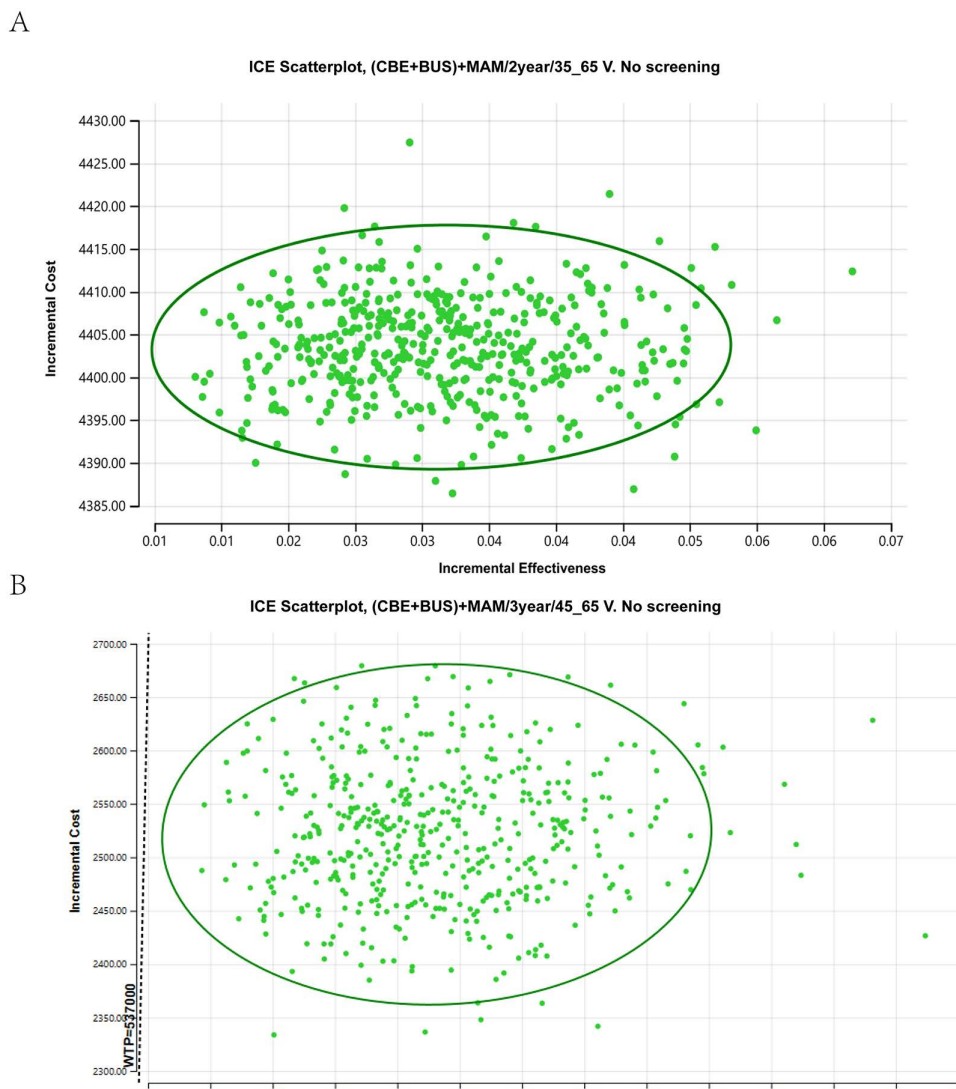

B

**Fig 7. A. Incremental cost-utilities scatter plot of probabilistic sensitivity analysis for (CBE + BUS)+MAM/2year/35_65 breast cancer screening vs no screening; B, Incremental cost-utilities scatter plot of probabilistic sensitivity analysis for (CBE + BUS)+MAM/3year/45_65 breast cancer screening vs no screening.** Each point in the scatter plot represents the result of one iteration from a second-order Monte Carlo simulation (1,000 iterations in total). Green points: iterations where the incremental cost-utility ratio (ICUR) falls below the willingness-to-pay (WTP) threshold of 537,000 CNY/QALY. The 95% confidence ellipse represents the joint uncertainty of the results, with its center at the mean incremental cost and QALYs, and its area encompassing approximately 95% of the simulated points.

when compared to other feasible alternatives? This study moves beyond merely validating the program's existence and seeks to identify the optimal screening strategy through a comprehensive cost-effectiveness analysis of 27 different protocols.

Our analysis first confirms that the current (CBE + BUS)+MAM/2year/35_65 strategy is unequivocally cost-effective compared to no screening, with an ICUR of 140,915.3 CNY/QALY, well below the Shenzhen-specific WTP threshold of 537,000 CNY/QALY. However, the more policy-relevant insight emerges from the incremental comparison among active strategies. The finding that the current strategy remains cost-effective (ICUR: 518,121 CNY/QALY) when directly

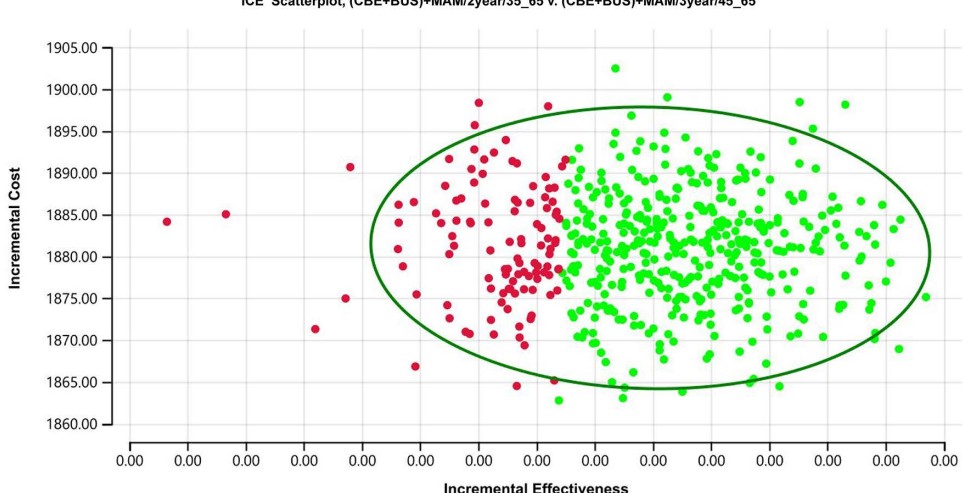

**Fig 8. Incremental cost-utilities scatter plot of probabilistic sensitivity analysis for (CBE+BUS)+MAM/2year/35_65 breast cancer screening vs (CBE+BUS)+MAM/3year/45_65.** Each point in the scatter plot represents the result of one iteration from a second-order Monte Carlo simulation (1,000 iterations in total), showing the incremental cost (ΔCost) and incremental quality-adjusted life years (ΔQALYs) of the (CBE+BUS) +MAM/2year/35_65 compared to (CBE+BUS)+MAM/3year/45_65. Green points: iterations where the incremental cost-utility ratio (ICUR) falls below the willingness-to-pay (WTP) threshold of 537,000 CNY/QALY. Red points: iterations where the ICUR exceeds the WTP threshold. The 95% confidence ellipse represents the joint uncertainty of the results, with its center at the mean incremental cost and QALYs, and its area encompassing approximately 95% of the simulated points.

compared to a plausible, less intensive alternative ((CBE+BUS)+MAM/3year/45_65) is particularly significant. It indicates that the additional resources required for initiating screening at age 35 and screening biennially, as opposed to starting at 45 triennially, are justified by the corresponding gains in quality-adjusted life years. This evidence solidifies the current protocol not just as a viable option, but as a strategically superior one within the context of Shenzhen's demographic and epidemiological profile. In contrast, other more intensive strategies (e.g., annual screening) were found not to be cost-effective, as their ICURs substantially exceeded the threshold.

The choice of imaging modalities requires regional adaptation. Mammography (MAM), while foundational in Western screening protocols, is less effective in dense breast tissue, a common characteristic among Asian populations [21, 33, 34]. In contrast, ultrasound (BUS) is more widely adopted in China but faces challenges in rural settings, where integrated CBE+BUS models have shown limited cost-effectiveness [35]. Shenzhen's approach, combining CBE with BUS and supplementary MAM, appears optimal for urban populations with comparable breast density and healthcare infrastructure.

Currently, the optimal starting age and ending age of breast cancer screening differ in different regions around the world, based on local female breast density and peak age of breast cancer incidence. The American Cancer Society recommends that women aged 45 and older undergo breast cancer screening, while those aged 40–44 have the option to start earlier. According to the 2016 guidelines from the U.S. Preventive Services Task Force (USPSTF), screening was recommended for women aged 50–74, with women aged 40–49 having the option to begin screening earlier. In the updated 2024 guidelines, the USPSTF expanded the recommended age range for screening to include women aged 40–74 [36–40]. In Asia, where breast tissue density is higher and peak incidence occurs earlier, countries like Singapore, Japan, and South Korea have adjusted their screening programs accordingly. Typically, these programs start at an age greater than 40 years [41–43]. As for the age distribution in this study, the age group of 40–44 years is the one that bore many breast cancer patients at 24.03%, followed closely by the 45–49 and 50–54 groups at 22.24% and 20.86%, respectively. There still is a good share of patients found in the 35–39 years age group, though—12.98%. For details, see Table 9. Cost-utility analysis

identifies (CBE + BUS)+MAM/2year/35_65 as the optimal strategy, with an ICUR of 140,915 CNY/QALY-well below the Shenzhen-specific WTP threshold of 537,000 CNY/QALY. The strategy is notable for their inclusion of an earlier starting age for screening, which is particularly relevant given the observed distribution of breast cancer cases across different age groups. Specifically, while the 40–44 years age group accounts for the highest proportion of cases (24.03%), a substantial portion of patients (12.98%) were also found in the 35–39 years age group. This finding underscores the importance of initiating screening at an earlier age to capture a significant number of potential cases.

Sensitivity analysis, together with Monte Carlo simulations, have been used to test the robustness of the screening model. In the current screening program (Fig 4), sensitivity analysis showed that the discount rate was the most influential aspect, as it is directly proportional to the time value in the case of cost-effectiveness analysis, thereby reflecting the present value calculation of future costs. Variations in the discount rate have a significant impact on the results of the cost-effectiveness of the entire model. The utility value for Stage IV breast cancer had the second largest impact, likely due to the high sensitivity of the model to late-stage breast cancer utility values. Treating late-stage breast cancer is more challenging and costly, so changes in utility values significantly influence model outcomes. The significant impact of ultrasound screening costs and follow-up expenses for stage IV cases underscores the importance of controlling the costs associated with screening technology and subsequent care in the actual screening process. This control is essential to ensure the economic sustainability of the screening program. The cost of mammography cannot be overlooked, as it plays a critical role in breast cancer screening, and its costs directly affect the overall economic burden of the screening process. From a cost-utility perspective, these breast cancer screening strategies may offer more economically effective health interventions for patients and society. Nonetheless, these conclusions need further validation in actual clinical settings and consideration of other potential influencing factors.

Despite the model' s strengths, several limitations warrant consideration. First, the assumption that all positive screens proceed to biopsy may overestimate real-world clinical pathways, as false positives can lead to unnecessary anxiety and downstream testing [44, 45]. However, evidence suggests that long-term psychological impacts of false positives are minimal, particularly among women over 40, and such results may even enhance health awareness and self-monitoring behaviors [45, 46]. Second, the risk of false negatives remains a concern, despite the high reliability of BI-RADS categories I and II. Improved imaging protocols, such as multimodal approaches combining ultrasound and mammography [35], along with enhanced clinician training, could mitigate this issue. Third, the study did not fully address the overdiagnosis paradox associated with ductal carcinoma in situ (DCIS). While DCIS detection has risen with widespread mammography adoption, its clinical significance remains debated, as not all lesions progress to invasive cancer [47]. Future research must prioritize developing precision tools to differentiate indolent from aggressive tumors, alongside refining risk stratification models to personalize screening intensity [48, 49]. Fourthly, our model assumes time-homogeneous transition probabilities between clinical stages, following the framework of Wong et al. [21]. While incidence and mortality are age-dependent, internal progression rates are constant. This may not fully capture variations in biological aggressiveness across age groups, a simplification necessitated by the scarcity of age-stratified progression data for Chinese women. Future studies with longitudinal data are needed to refine these probabilities. While Markov decision models are theoretically sound, their calibration to real-world scenarios demands continuous data collection through long-term follow-up.

## Conclusion

This study demonstrates that the current Shenzhen breast cancer screening strategy ((CBE + BUS) +MAM/2year/35_65) is not only a cost-effective investment compared to no screening but, more critically, represents the optimal choice among a spectrum of 27 alternative strategies. The finding that this strategy remains cost-effective when directly compared to a less intensive alternative ((CBE + BUS) +MAM/3year/45_65), with an ICUR of 518,121 CNY/QALY, provides robust evidence to sustain the existing program. It justifies the policy decision to initiate screening at age 35 and conduct it biennially as a cost-effective balance between maximizing early detection and ensuring resource efficiency within Shenzhen's specific socioeconomic context.

## Supporting information

**S1 Table. Cost-Effectiveness Rankings.**
(XLSX)

**S2 Table. Breast Cancer Screening Strategies.**
(XLSX)

**S3 Table. Distribution parameters for cost of screening program.**
(XLSX)

**S4 Table. Distribution parameters for medical costs.**
(XLSX)

**S5 Table. Distribution parameters for health utility value.**
(XLSX)

**S6 Table. Distribution parameters for sensitivity and specificity.**
(XLSX)

**S1 Fig. Supplementary.**
(DOCX)

**S1 Appendix. Supplementary Appendix 1.**
(DOCX)

## Acknowledgments

The authors acknowledge the Shenzhen Maternity and Child Healthcare Hospital for providing the breast cancer screening data. We are grateful for the support of the healthcare professionals involved in the screening program and the contributions of the participants. We thank Chen WT for assistance with language editing and manuscript formatting.

## Author contributions

**Conceptualization:** XueSen He.

**Data curation:** Changqing Tu, Yuke Zhong, Haifeng Qi.

**Formal analysis:** Changqing Tu, Yuke Zhong, Qian Lu.

**Investigation:** XueSen He.

**Methodology:** Changqing Tu, Huan Li, Haifeng Qi, Qian Lu, XueSen He.

**Resources:** Changqing Tu.

**Software:** Changqing Tu, Yuke Zhong, Huan Li, Haifeng Qi, Qian Lu.

**Supervision:** XueSen He.

**Validation:** Changqing Tu, Huan Li.

**Visualization:** Changqing Tu.

**Writing – original draft:** Changqing Tu, XueSen He.

**Writing – review & editing:** Changqing Tu, Yuke Zhong, XueSen He.

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
