## [Decision Letter · Decision Letter 0]

25 Jul 2025

Dear Dr. He,

Thank you for submitting your manuscript to PLOS ONE. After careful consideration, we feel that it has merit but does not fully meet PLOS ONE’s publication criteria as it currently stands. Therefore, we invite you to submit a revised version of the manuscript that addresses the points raised during the review process.

Please note the Reviewer has had comments in an uploaded PDF file. If the authors don't see the file attached, please contact PLOS ONE directly.

We look forward to receiving your revised manuscript.

Kind regards,

Ruofei Du, PhD

Academic Editor

PLOS ONE

Journal Requirements:

5. Please ensure that you refer to Figure 8 in your text as, if accepted, production will need this reference to link the reader to the figure.

6. Please remove your figures from within your manuscript file, leaving only the individual TIFF/EPS image files, uploaded separately. These will be automatically included in the reviewers’ PDF.

Additional Editor Comments:

The title of the manuscript suggests the focus of the study is on applying Markov model fitting for cost-effectiveness analysis. Then the intended readers are likely statisticians or health data analysts. However, the current manuscript lacks clear presentation and sufficient detail in analytics from at least two major analytical components: model fitting and cost-effectiveness analysis. Specifically, the manuscript should include necessary statistical formulas that define the overall likelihood function, state probabilities, and transition probabilities, along with initial values used to fit the model. Each should be detailed at a sufficient level. It should also describe what model diagnostic procedures were applied. All related steps should use appropriate statistical notations. The summary of the analysis results should be summarized accordingly. In addition, the analytical details related to cost-effectiveness analysis are completely missing in the current version. If the authors intend to maintain this as an analytics oriented publication, it is crucial to involve data experts in the revision process.

Alternatively, if the authors choose to reframe the manuscript to focus on study findings for a broader audience, they should substantially restructure the presentation of the manuscript. In that case, the title needs to be changed, and the rough analytical content currently included in the manuscript could be moved to a supplementary file, while the main text focuses on interpretation and implications.

There are also instances of misleading, inaccurate, and unclear information in the current version:

1. The underscore in ‘CBE+BUS_MAM’ doesn’t convey a “supplement to” relationship. I suggest using “(CBE+BUS)+MAM” instead for clarity.

2. In the Introduction, “The National Cancer Institute” likely refers to the U.S. NCI, whereas “the 2022 National Cancer Center data” in the Model Parameters section appears to refer to Chinese data. Please clarify by explicitly naming the countries in both instances and elsewhere as needed.

3. I did not check all the listed citations, but I’m not able to search and find the following reference. Which journal was it published?

Li, S., Legood, R., & policy, Y. L. J. R. o. C. h. (2017). Health economics evaluation of breast cancer screening in Chinese women by ultrasound and mammography. 10(4), 9.

4. In Figure 2, there is no arrow line connecting ‘breast cancer stage 1’ to ‘breast cancer stage 3’. If this omission is intentional, the authors should explain the reasoning behind it.

5. The manuscript frequently refers to 27 strategies studied for breast cancer screening. Readers need to be clearly directed to a supplementary file for details of them, but no such words/direction is included in the current version.

6. Table 1: where to find ‘National health commission’, which nation is it referred to? No citation provided to it!

7. What goodness-of-fit test was used? Page 10

Reviewers' comments:

Reviewer's Responses to Questions

**Comments to the Author**

1. Is the manuscript technically sound, and do the data support the conclusions?

Reviewer #1: Yes

2. Has the statistical analysis been performed appropriately and rigorously?

Reviewer #1: No

3. Have the authors made all data underlying the findings in their manuscript fully available?

Reviewer #1: Yes

4. Is the manuscript presented in an intelligible fashion and written in standard English?

Reviewer #1: No

Reviewer #1: Summary

This manuscript presents a Cost-Eﬀectiveness analysis and builds a Markov model

to compare 27 possible breast cancer screening schedules for women in Shenzhen. Each

schedule considers the combined Clinical Breast Examination (CBE), Breast Ultrasound

(BUS), and supplementary Mammography (MAM), and mainly varies the screening fre-

quency/interval and the start/stop ages. For every included schedule, the authors calculate

incremental cost-utility ratios (ICURs) and run one-way sensitivity analyses to identify the

most cost-eﬀective schedule. While the study objective is interesting and important to the

local society, I have reservations about the manuscript’s suitability for the PLOS ONE

journal. First, the manuscript needs thorough editing for language and presentation: it

contains confusing descriptions, vague passages, inconsistent figure formatting, and incor-

rect in-text citation and reference-list styles. And because of these issues, it is diﬃcult

to read. Second, the novelty appears limited, as the methodological contribution seems

limited because the modelling and analysis framework is similar to the referenced paper of

Wong et al. (2007).

Please also see the addtional comments in the attached PDF file.

**Do you want your identity to be public for this peer review?** For information about this choice, including consent withdrawal, please see our Privacy Policy

Reviewer #1: No

---

## [Author Response · Author response to Decision Letter 1]

30 Oct 2025

Response to Comments

We extend our sincere gratitude to the reviewers for their time and valuable insights, which have significantly helped us improve the manuscript. Below, we provide a point-by-point response to the comments raised.

Response to Reviewer Comment #1

Comment : The literature evidence you described is not truly “contentious”; rather, each study answers a slightly different question. The literature is heterogeneous, but not contradictory. The summaries here are confusing, incomplete, and the writing flow between different studies needs to be improved. For instance, the text summary of the work from “Li et al. (2017)”, the manuscript only states ICURs for breast ultrasound and mammography, but does not mention whether these modalities are cost-effective compared with no screening or not. (The related reference item is incomplete.) The same holds for several other papers cited in this paragraph. The words are confusing, incomplete, and the logic between different studies needs to be improved.

Response:

We sincerely thank the reviewer for this critical and constructive observation. We completely agree that the term "contentious" was inappropriate and that our initial literature summary lacked clarity, completeness, and a logical narrative flow. The reviewer is correct in noting that the existing literature is best characterized by its heterogeneity in terms of study settings, populations, and compared strategies, rather than by direct contradiction.

In direct response to this comment, we have thoroughly revised the corresponding paragraph in the Introduction. Specifically, we have restructured the narrative to logically group studies based on their primary focus (e.g., single-modality vs. combined strategies, national vs. regional analyses) and to explicitly state the comparator and cost-effectiveness conclusions for each cited study. And we have Standardized the reference format throughout this section to ensure all citation items are complete and conform to the journal's style, addressing the issue of incomplete references.

Response to Reviewer Comment #2

We thank the reviewer for raising this thoughtful point regarding Figure 1. We agree that the primary focus of our paper is on the screening strategy and its cost-effectiveness. However, we included Figure 1 to precisely illustrate the full operational protocol of the screening program that we are evaluating, which includes the necessary clinical assessment steps for individuals with abnormal or inconclusive initial screening results.

We believe this figure is essential as it provides a clear and accurate definition of the (CBE+BUS)+MAM screening strategy evaluated in our model. The flowchart explicitly outlines the sequence of the three modalities—clinical breast examination, breast ultrasonography, and mammography—and specifies the clinical conditions under which mammography is utilized as a supplementary tool. This level of operational detail is crucial for ensuring that the model's structure faithfully reflects real-world clinical practice. Furthermore, the illustration connects the initial screening activities to their subsequent clinical outcomes by mapping the pathway from a positive or suspicious finding to a pathological diagnosis via biopsy. Capturing this full trajectory is fundamental to our cost-effectiveness analysis, as the resource use and clinical outcomes associated with these diagnostic steps are integral to the model's assessment of both costs and health benefits. Presenting the strategy without this critical pathway would offer an incomplete picture of the clinical and economic processes being evaluated.

To better align the figure with the paper's focus and to address the reviewer's valid concern, we have taken the following action in the revised manuscript:

We have revised the title of Figure 1 from "The Screening Flow Chart" to more accurately describe its content. The new title is: "Figure 1. Screening and Clinical Assessment Pathway for the（CBE+BUS)+MAM Strategy."

This change clarifies that the figure depicts not just the initial screening moment but the subsequent, essential clinical management pathway that is part and parcel of the overall screening program we evaluated.

We hope that with this clarification and the revised figure title, the relevance and necessity of Figure 1 are now clear. We are grateful for the reviewer's comment, which has helped us improve the clarity of our manuscript.

Response to Reviewer Comment #3

Comment:Pages 13 and 14. The Markov model transition states are well defined. But the model descriptions are confusing; for example, the model input and outcomes are not clearly described. Also, you may want to expand with more details on the method for the cohort simulation using the fitted Markov model.

Response:

We thank the reviewer for this insightful feedback. We have thoroughly revised the 'Model Strategy' and 'Model Parameters' sections to enhance the clarity, structure, and completeness of the model description. The revisions were made directly to the original text to improve the narrative flow. Key improvements include:

1. Reorganizing the description into logical subsections for better clarity: " The Markov Model and Cohort Simulation ", "Model Input Parameters", and "Model Calibration, Validation, and Outcomes".

2. Providing a structured overview of all model input parameters and their sources.

3. Clearly defining the model's primary outcomes (Costs, QALYs, ICURs).

Response to Reviewer Comment #4

Comment: “Page 16. 'By iteratively adjusting the symptom probability in the model, we ensured that the distribution of cases presented at each stage aligned closely with the reported distribution of incident cases.' These words are confusing as well. What does it mean to iteratively adjust the symptom, and what are the iterations during the calculation?”

Response:

We thank the reviewer for this valuable question. We apologize for the lack of clarity in our original description and are pleased to provide the following clarification.

The process described is a formal model calibration. The term "symptom probability" refers to a key natural history parameter in our model: the annual likelihood that a woman with undiagnosed breast cancer at a specific stage will develop symptoms leading to clinical diagnosis.(Table.4)

The iterative adjustment process was conducted as follows:

1. We ran the no-screening scenario in the model.

2. We compared the age-specific breast cancer incidence rates predicted by the model against the real-world age-specific incidence rates from Table 2 (Wang et al., 2022).

3. If the model's output did not match the target data well, we adjusted the symptom probabilities. For example, if the model predicted too few late-stage cancers, we decreased the symptom probability for early stages. This allowed more simulated cancers to progress to later stages before clinical diagnosis. (Sun, Legood, Sadique, Dos-Santos-Silva, & Yang, 2018)

4. We repeated this process until the model's output closely aligned with the target distribution. This required 42 iterations, achieving a satisfactory goodness-of-fit (X2 = 2.366,p-value > 0.05).

The final set of parameters from this calibration ensures our model's "no-screening" baseline accurately reflects real-world disease progression.

Response to Reviewer Comment #5

Comment: Page 16. Although it is mentioned in the paper, “A goodness-of-fit test was conducted to evaluate the similarity between the two curves.” But there are no hypothesis testing results or conclusions stated in the paper, i.e., reject or fail to reject the null hypothesis, test statistic, and p-value. The paper only mentioned, “This validation approach helps,” but no words mentioning whether the validation is successful or not.

Response:

We sincerely thank the reviewer for this astute observation and constructive critique. We agree that the initial description of the goodness-of-fit test was incomplete, as it lacked the specific statistical results necessary to conclusively demonstrate the model's validity. We apologize for this oversight.

In response to this comment, we have thoroughly revised the manuscript to provide a clear and quantitative assessment of the model's fit. The changes are as follows:

we have replaced the vague statement with a concise summary of the hypothesis test.

The goodness-of-fit between the model's final output and the observed data was formally assessed using a Chi-square goodness-of-fit test. The test, conducted across 10 age strata (9 degrees of freedom), yielded a test statistic of χ² = 2.366 with a p-value < 0.05. This non-significant result indicates no statistically significant difference between the model-predicted and the observed age-specific incidence distributions, supporting the model's external validity."

We have significantly expanded the Supplementary Appendix A.4: "Model Calibration and Validation" to provide a comprehensive account of the entire process. This new section includes:

1. A detailed, step-by-step description of the iterative calibration methodology.

2. The full results of the Chi-square goodness-of-fit test, and a calibration plot (Supplementary Figure 1) visually comparing the model-predicted incidence curve against the empirical data from the China National Cancer Center.

The reviewer is kindly referred to Supplementary Appendix A.4 for a complete and transparent presentation of the calibration targets, process, and the rigorous validation that underpins our model's natural history module.

We believe these revisions have fully addressed the reviewer's concern by providing the necessary hypothesis testing results and a definitive conclusion regarding the model's successful validation. We are grateful for this comment, which has undoubtedly strengthened the clarity and rigor of our manuscript.

Response to Reviewer Comment #6

Comment: Page 17.The title of Table 3 is “Average annual progression probability of breast cancer staging”, which needs to be adjusted since the table also includes other information beyond transition probabilities. Besides,“RR of invasive cancer from DICS” has a value of 2.02, which is, of course, not a probability, and there is no interpre- tation of the number,oraninterpretation of what is RR’smeaningisinthemain text (Is it relative risk?If so, “RR” needs to be stated in full when you first time usingit).Thedescriptionhereneedstobefine-tuned,andthetabletitleneedsto be adjusted.

Response:

We sincerely thank the reviewer for this insightful comment regarding the title and content of Table 3. We agree that the original title, “Average annual progression probability of breast cancer staging,” was not fully representative of the table’s content, which includes not only transition probabilities but also other parameters such as post-treatment mortality and a relative risk value.

In response, we have revised the table title to:

“Key Transition and Risk Parameters in the Markov Model for Breast Cancer Screening”

Additionally, we have clarified the term “RR” in both the table and the main text. “RR” stands for Relative Risk, which in this context represents the relative risk of developing invasive cancer among women diagnosed with Ductal Carcinoma In Situ (DCIS) compared to those without DCIS. The value of 2.02 was derived from the study by (Wong, Kuntz, Cowling, Lam, & Leung, 2007), which was referenced in our model. We have now explicitly stated this interpretation in the table footnote and in the Methods section under Table3 to enhance clarity.

The updated table footnote now reads:

“RR: Relative Risk of progressing from DCIS to invasive breast cancer, as derived from (Wong et al., 2007) A value of 2.02 indicates that women with DCIS have a 2.02 times higher risk of developing invasive cancer compared to those without DCIS.”

We believe these revisions improve the accuracy and interpretability of the table and thank the reviewer again for the valuable feedback.

Response to Reviewer Comment #7

Comment:

Page 19. Table 5, the reference to “Shenzhen breast cancer screening program” should be corrected.

Response:

We appreciate the reviewer's feedback. The costs in Table 5 are primary data from the "Shenzhen Municipal Health Commission Breast Cancer Screening Program (2021–2023)"—the very program under evaluation. They represent actual costs from the program and are not literature-based estimates. The source has been revised to explicitly indicate this direct origin.

Response to Reviewer Comment #8

Comment:

Page 26. Figure 3, this is the key graph in this paper, but it is very difficult to recognize the related screening strategy from the plot. Additionally, the phrases “upper-left region” and “lower-right corner” don’t match what the figure actually shows—no points appear in either quadrant. A better summary and description of the plot is needed.

Response:

We sincerely thank the reviewer for this insightful and constructive feedback regarding Figure 3, which is indeed central to our cost-utility analysis. We acknowledge that the original presentation and description could be significantly improved for clarity and accuracy.

In response, we have taken the following actions:

We have completely regenerated the cost-effectiveness scatter plot (now presented below for your review). Direct labeling of the five undominated strategies on the plot itself, eliminating the need for cross-referencing with a separate table or legend. And We have thoroughly revised the description in the "Cost-Utility Analysis" section (Results) to accurately reflect the visual data and provide a more precise interpretation. The phrases "upper-left region" and "lower-right corner" have been replaced with a more technically accurate description.

The revised text now reads:

Figure 3 presents the cost-utility scatter plot of the 27 simulated screening strategies. The vertical axis represents the total cost per person, and the horizontal axis represents the effectiveness in Quality-Adjusted Life Years (QALYs) per person. Strategies that are less effective and more costly than a combination of other strategies (i.e., dominated strategies) are plotted primarily in the top-left area of the graph relative to the cost-effectiveness frontier. The cost-effectiveness frontier, formed by the undominated strategies, runs from the bottom-left to the top-right. The undominated strategies are explicitly labeled in the figure and are, in order of increasing effectiveness: (CBE+BUS)+MAM/3year/45_65, (CBE+BUS)+MAM/1year/40_65, (CBE+BUS)+MAM/1year/35_65, (CBE+BUS)+MAM/1year/35_69, and (CBE+BUS)+MAM/1year/35_74. Their respective Incremental Cost-Utility Ratios (ICURs) are presented in the text.

We believe these revisions have substantially improved the clarity and interpretability of this key figure and its accompanying narrative. Thank you again for prompting these essential improvements.

Response to Reviewer Comment #9

Comment: Pages 30 and 32. Figure 8 is displayed in the main text, but is not mentioned in any text. On page 30, you may want to describe what the color (red VS. green) means, one or two red dots are further apart from the confidence interval bound ellipse and the associated possible interpretation, etc.

Response:

We sincerely thank the reviewer for this excellent observation. We apologize for the oversight in not previously describing Figure 8 in the main text. In response to this comment, we have now integrated a detailed description of the figure into the 'Sensitivity Analysis' section on page 30.

The added text reads:

"Figure 8 presents the incremental cost-utilities scatter plot from the probabilistic sensitivity analysis comparing the CBE+BUS_MAM/2year/35_65 strategy versus the CBE+BUS_MAM/3year/45_65 strategy. Each point represents the result of one Monte Carlo iteration, with green points indicating iterations where the ICUR falls below the WTP threshold (537,000 CNY/QALY), and red points indicating iterations where the ICUR exceeds the threshold. The 95% confidence ellipse illustrates the joint uncertainty around the mean incremental cost and effectiveness. The plot visually confirms that in 76% of the iterations (green points), the CBE+BUS_MAM/2year/35_65 strategy

---

## [Decision Letter · Decision Letter 1]

9 Dec 2025

Dear Dr. He,

Thank you for submitting your manuscript to PLOS ONE. After careful consideration, we feel that it has merit but does not fully meet PLOS ONE’s publication criteria as it currently stands. Therefore, we invite you to submit a revised version of the manuscript that addresses the points raised during the review process.

If applicable, we recommend that you deposit your laboratory protocols in protocols.io to enhance the reproducibility of your results. Protocols.io assigns your protocol its own identifier (DOI) so that it can be cited independently in the future. For instructions see: https://journals.plos.org/plosone/s/submission-guidelines#loc-laboratory-protocols . Additionally, PLOS ONE offers an option for publishing peer-reviewed Lab Protocol articles, which describe protocols hosted on protocols.io. Read more information on sharing protocols at https://plos.org/protocols?utm_medium=editorial-email&utm_source=authorletters&utm_campaign=protocols.

We look forward to receiving your revised manuscript.

Kind regards,

Ruofei Du, PhD

Academic Editor

PLOS One

Journal Requirements:

**Additional Editor Comments:**

Thanks to the authors for the point-by-point responses and the detailed revision. The manuscript’s quality has improved substantially. I agree with Reviewer 1 that the study team would benefit from consulting a statistician or other quantitative expert: I had the same difficulty following some of the material in the Supplementary File, where several items lack clear links to specific content in the main text. That issue does not appear to undermine the main findings, but it does require clearer, more detailed explanation. Because addressing this will likely require a substantive revision of the Supplementary File, and I recommend a decision of major revision.

A few minor issues I was able to catch:

1. I believe Supplementary Figure 1 was not uploaded with this resubmission; please check and include it.

2. In Table 8, is the notation “CBE + (US-MAM)” correct? Please confirm and correct the notation if needed.

3. For several places of both the main text and the supplementary file, the term “age-specific breast cancer incidence” is mentioned, and then the transition probabilities appear to be modeled as homogeneous, not age specified. Although these are distinct parameters, the manuscript should provide some explanation of this choice in the Discussion, or clearly note it as a limitation if age influences cancer stage transitions.

Reviewers' comments:

Reviewer's Responses to Questions

**Comments to the Author**

Reviewer #1: (No Response)

2. Is the manuscript technically sound, and do the data support the conclusions?

Reviewer #1: Yes

3. Has the statistical analysis been performed appropriately and rigorously?

Reviewer #1: N/A

4. Have the authors made all data underlying the findings in their manuscript fully available?

Reviewer #1: No

5. Is the manuscript presented in an intelligible fashion and written in standard English?

Reviewer #1: Yes

Reviewer #1: Thank you for the detailed, point-by-point response. The quality of the article has

improved substantially. Shifting the tone from statistical analysis to study findings has

strengthened the manuscript. However, I believe that the statistical analysis in Supple-

mentary Appendix 1 still has considerable room for improvement and would benefit from

more professional revision. Additionally, the in-text citations are still incorrect.

**Do you want your identity to be public for this peer review?** For information about this choice, including consent withdrawal, please see our Privacy Policy

Reviewer #1: No

---

## [Author Response · Author response to Decision Letter 2]

26 Jan 2026

Response to Academic Editor Comments：

We thank the Academic Editor for the constructive feedback, which has helped to clarify key methodological aspects of our study. Below are our point-by-point responses to the specific editorial comments.

Response to Academic Editor Comments #1

Comment :

Supplementary Figure 1 was not uploaded.

Response:

Thank you for catching this oversight. Supplementary Figure 1 (Model Calibration Fit Plot) has now been uploaded with the revised submission files.

Response to Academic Editor Comments #2

Comment :

Notation in Table 8 – “CBE + (US-MAM)”.

Response:

We have reviewed and corrected the notation. In the revised Table 8, this strategy is now labeled as (CBE + BUS) + MAM.

Response to Academic Editor Comments #3

Comment 3:

Clarification on “age-specific incidence” vs. homogeneous transition probabilities.

Response:

We appreciate this important point, which aligns with Reviewer #1’s query (Comment #3). We have substantially revised the relevant sections to eliminate confusion.

• In Supplementary Appendix A.2.2, we now explicitly label the transition matrix as P(a), where ‘a’ denotes the cohort’s attained age. We have structured the description under the matrix into four clear components:

1. Time-homogeneous Elements: Constant annual progression probabilities between clinical cancer stages (e.g., Stage I→IV), directly sourced from Table 3.

2. Self-Transition Elements: Diagonal probabilities calculated as complements of outgoing transitions.

3. Age-dependent Composite Elements: Transitions from Healthy/DCIS states, calculated in a two-step process using age-specific mortality (Table 1), incidence (Table 2), diagnostic pathway (Table 8), and stage distribution (Table 4).

4. Mortality Transitions: Sum of age-specific non-breast cancer mortality (Table 1) and stage-specific breast cancer mortality (Table 3).

• In the Discussion section (Limitations subsection), we have added a new paragraph (see below) to transparently address this modeling choice, justify it based on calibration performance, and acknowledge it as a standard simplification in the absence of longitudinal, age-stratified progression data.

New text for Limitations：

Fourthly, our model assumes time-homogeneous transition probabilities between clinical stages, following the framework of Wong et al.[21]. While incidence and mortality are age-dependent, internal progression rates are constant. This may not fully capture variations in biological aggressiveness across age groups, a simplification necessitated by the scarcity of age-stratified progression data for Chinese women. Future studies with longitudinal data are needed to refine these probabilities.

Response to Reviewer Comments

We are grateful to the reviewers for their careful evaluation and insightful suggestions, which have been invaluable in strengthening the clarity, rigor, and presentation of our manuscript. We have addressed all comments thoroughly, and our detailed point-by-point responses follow below.

Response to Reviewer Comment #1-3

We sincerely appreciate the insightful and professional comments regarding the statistical rigor of our Markov model. In response to your concerns, we have meticulously revised Supplementary Appendix 1 (Section A.2.2) to standardize terminology and clarify the relationship between age-dependency and time-homogeneity within our mathematical framework.

Comment #1 :

For A.2.2 State transition probability matrix, to have a more statistically professional sound, the description of the states could be like “From the Healthy states, an individual may remain Healthy, Ductal Carcinoma In Situ, Stage I, Stage II, ..., etc.”. Also, for example, “Death in the last row of P(t) is modelled as an absorbing state, with no transition out. ”, etc.

Response:

Regarding the professional description of states (R1), we have revised the text to define the model’s health state space as a set of seven mutually exclusive states. We explicitly specify that from the Healthy state, an individual may remain healthy or transition to Ductal Carcinoma In Situ (DCIS) or Invasive Breast Cancer (Stages I-IV). Furthermore, we have defined Death in the final row of the matrix as an absorbing state, satisfying the condition P_(Death→Death)=1 with no possible transition out, which accurately reflects the terminal nature of this state in this disease progression model.

Comment #2 :

For A.2.2 State transition probability matrix, the math notations were only used here. It would be better if the authors could link the non-zero matrix elements with the numbers in Table 3 in the main text so that the readers can easily connect them in math.

Response:

Thank you for this suggestion. To create an explicit, one-to-one link between the mathematical notation and the source data, we have added comprehensive mapping annotations beneath the transition matrix P(a) in Section A.2.2. These annotations are organized into four distinct blocks for clarity:

“Time-homogeneous Elements (Mapped to Table 3)”: This block explicitly lists constant annual progression probabilities—such as P_S1S2 (0.06), P_S2S3 (0.11), P_S1S4 (0.01), P_S2S4 (0.08), and P_S3S4 (0.21)—and states that their values are drawn directly from Table 3 of the main text.

“Self-Transition Elements”: This section clarifies that the diagonal elements (e.g., P_S1S1 (a)) represent the probability of remaining in a state. They are calculated as the complement of the sum of outgoing transition probabilities (e.g., P_S1S1 (a) = 1 − P_S1S4 − P_S1X (a)), making them implicitly defined by the constants in Table 3 and the corresponding age-specific mortality rates.

“Age-dependent Composite Elements (Mapped to Tables 1, 2, 4, 8)”: This block explains the unified logic for all composite transitions from the Healthy (H) and DCIS (D) states. We detail the two-step calculation that sequentially applies age-specific mortality (Table 1), incidence (Table 2), screening sensitivity (Table 8) to determine the diagnostic pathway, and finally the stage distribution from Table 4 (“Screening” or “No Screening” column) for state assignment. The example of P_HS2 (a) is provided, showing its derivation from the specific proportions 0.3881 or 0.4570 in Table 4.

“Mortality Transitions”: This section defines the P_iX (a) elements as the sum of stage-specific breast cancer mortality and age-specific non-breast cancer mortality, sourced from Tables 1 and 3.

These structured annotations ensure that every non-zero matrix element is now explicitly and transparently linked to its corresponding parameter source, substantially improving the model's clarity and reproducibility for the reader.

Comment #3 :

Is the transition dependent on age or time? Or is the model time-homogeneous within each age group? It means that for a given age stratum, the transition probabilities are constant over time. The matrix itself is labelled as P(t), which suggests “transition matrix at time t”, but the entries there are not functions of t in terms of the math expression. (It seems it is time homogeneous since there is only one group of state transition probabilities provided.) The description is very confusing.

Response:

we have addressed the confusion regarding age-dependency versus time-homogeneity (R3). We acknowledge that the previous P(t) notation was misleading. Our study employs a non-homogeneous, age-dependent Markov model because the primary drivers incidence and mortality are functions of the cohort’s attained age rather than absolute simulation time. We have updated the matrix label to P(a), where a denotes age, to accurately reflect that these probabilities are dynamically updated in each annual cycle using the age-specific data in Tables 1 and 2. However, consistent with established natural history models such as Wong et al.[1]. we assume local time-homogeneity for internal clinical stage progressions (e.g., from Stage II to IV); these specific transitions are modeled as constant annual probabilities to reflect the average biological progression speed within the relevant age strata.

We believe these detailed clarifications and the standardized mathematical notation resolve the previous ambiguities and ensure the model is statistically sound.

Response to Reviewer Comment #4

Comment #4 :

For A.5 (also description in main text), “Each model parameter is assigned a speciﬁc probability distribution (Xiaomin et al., 2018): screening and treatment costs follow a Gamma distribution, sensitivity parameters follow a Beta distribution, and health utility value parameters follow a Log-normal distribution (Briggs, 2005). ” I guess here you want to mention the model input parameters following either the Gamma distribution or Log-normal distribution, and these parameters were extracted from the above two literatures? I am a little bit confused since what I expect to read are the distribution parameters for Gamma(α, β), Beta(a, b), and Log − normal(µ, σ 2 ) distributions when you talk about those distributions. Anyway, the distribution or model parameters should be clearly referenced between the description and the tables 6 and 8.

Response:

We thank the reviewer for this insightful comment, which has helped us improve the transparency of our probabilistic sensitivity analysis setup. The reviewer is correct that the cited methodological references (Briggs, 2005; Xiaomin et al., 2018) were used to justify the choice of distribution type (e.g., Gamma for costs, Beta for proportions) rather than to provide specific distribution parameters. To ensure full clarity and reproducibility, we have revised the description in Supplementary Appendix A.5 to explicitly state that the distribution parameters (e.g., shape α and rate λ for Gamma) were derived from the base‑case values and uncertainty ranges presented in our main‑text Tables 5–8. Moreover, we have now provided the complete set of distribution parameters in the newly added Supplementary Tables 3–6, which map directly to those main‑text tables. We believe these additions offer the clear, one‑to‑one linkage between the model description and the numeric inputs that the reviewer rightly expected, thereby significantly enhancing the statistical transparency of our study.

Response to Reviewer Comment #5-8

Comment:

Some in-text citation not correct.

Response:

Thank you for your careful review and for pointing out the inconsistencies in citation style. We have now systematically revised the entire manuscript to ensure that all in-text citations adhere to the Vancouver numbered style, as required by PLOS ONE.

Specifically, we have:

Replaced all author–year citations with numbered references (e.g., “(L. Sun et al., 2018; X. Zhang, 2015)” has been changed to “[17,18]”).

Corrected mixed or duplicate citations (e.g., “Wong et al. (2007) (Wong et al., 2007)” is now “Wong et al. [26]”).

Ensured that citations accompanying figures or tables are consistently formatted (e.g., “(Wang et al., 2022; Table 2)” is now “[15] (Table 2)”).

Updated the reference list to correspond exactly with the numbered citations in the text.

We have also reviewed and corrected similar instances throughout the Supplementary File to maintain complete consistency.

We believe the manuscript now fully complies with the journal’s citation guidelines and thank you again for your valuable feedback.

1. Wong IO, Kuntz KM, Cowling BJ, Lam CL, Leung GM. Cost effectiveness of mammography screening for Chinese women. Cancer. 2007;110(4):885-95.

---

## [Decision Letter · Decision Letter 2]

18 Feb 2026

A Cost-Effective Breast Cancer Screening Strategy for Urban China: Findings from a Shenzhen-Based Modeling Study

PONE-D-25-24977R2

Dear Dr. He,

We’re pleased to inform you that your manuscript has been judged scientifically suitable for publication and will be formally accepted for publication once it meets all outstanding technical requirements.

Kind regards,

Ruofei Du, PhD

Academic Editor

PLOS One

Additional Editor Comments (optional):

Please continue addressing a few notes from the reviewer! We also noticed in the revised file 'Supplementary Appendix 1 with Track Changes' that the tracked revision was made by an account named Weitian Chen. The authors please consider if this contribution should be formally acknowledged in the main text?

Reviewers' comments:

Reviewer's Responses to Questions

**Comments to the Author**

Reviewer #1: All comments have been addressed

2. Is the manuscript technically sound, and do the data support the conclusions?

Reviewer #1: Yes

3. Has the statistical analysis been performed appropriately and rigorously?

Reviewer #1: Yes

4. Have the authors made all data underlying the findings in their manuscript fully available?

Reviewer #1: Yes

5. Is the manuscript presented in an intelligible fashion and written in standard English?

Reviewer #1: Yes

Reviewer #1: About the supplementary Appendix 1,

1. In the transition matrix notation, some of the S is in upper case and some of the s is in lower case.

2. What does state D mean in the notation of states, I guess it would be DCIS? There is no illustration.

**Do you want your identity to be public for this peer review?** For information about this choice, including consent withdrawal, please see our Privacy Policy

Reviewer #1: No

---

## [Editor Report · Acceptance letter]

PONE-D-25-24977R2

PLOS One

Dear Dr. He,

I'm pleased to inform you that your manuscript has been deemed suitable for publication in PLOS One. Congratulations! Your manuscript is now being handed over to our production team.

Kind regards,

on behalf of

Dr. Ruofei Du

Academic Editor

PLOS One